# A Novel Method for Identifying the Transition Zone in Long-Segment Hirschsprung Disease: Investigating the Muscle Unit to Ganglion Ratio

**DOI:** 10.3390/biom12081101

**Published:** 2022-08-10

**Authors:** Wendy Yang, Jenny Pham, Sebastian K. King, Donald F. Newgreen, Heather M. Young, Lincon A. Stamp, Marlene M. Hao

**Affiliations:** 1Department of Anatomy and Physiology, The University of Melbourne, Parkville 3010, Australia; 2Graduate Institute of Clinical Medicine, College of Medicine, National Taiwan University, Taipei 10663, Taiwan; 3Murdoch Children’s Research Institute, Parkville 3052, Australia; 4Department of Paediatric Surgery, The Royal Children’s Hospital, Parkville 3052, Australia; 5Department of Paediatrics, The University of Melbourne, Parkville 3010, Australia

**Keywords:** Hirschsprung disease, enteric nervous system, transition zone, enterocolitis

## Abstract

Hirschsprung disease (HSCR) is characterised by the absence of enteric ganglia along variable lengths of the distal bowel. Current gold standard treatment involves the surgical resection of the defective, aganglionic bowel. Clear and reliable distinction of the normoganglionated bowel from the transition zone is key for successful resection of the entire defective bowel, and the avoidance of subsequent postoperative complications. However, the intraoperative nature of the tissue analysis and the variability of patient samples, sample preparation, and operator objectivity, make reproducible identification of the transition zone difficult. Here, we have described a novel method for using muscle units as a distinctive landmark for quantifying the density of enteric ganglia in resection specimens from HSCR patients. We show that the muscle unit to ganglion ratio is greater in the transition zone when compared with the proximal, normoganglionated region for long-segment HSCR patients. Patients with short-segment HSCR were also investigated, however, the muscle unit to ganglion ratio was not significantly different in these patients. Immunohistochemical examination of individual ganglia showed that there were no differences in the proportions of either enteric neurons or glial cells through the different regions of the resected colon. In addition, we identified that the size of enteric ganglia was smaller for patients that went on to develop HSCR associated enterocolitis; although the density of ganglia, as determined by the muscle unit to ganglia ratio, was not different when compared with patients that had no further complications. This suggests that subtle changes in the enteric nervous system, even in the “normoganglionated” colon, could be involved in changes in immune function and subsequent bacterial dysbiosis.

## 1. Introduction

The enteric nervous system (ENS) is a complex network of neurons and glia in the gastrointestinal tract that regulates its motility, secretory, absorptive and immune functions [1]. In humans, it consists of approximately 500 million neurons, distributed between the myenteric plexus and the submucosal plexus. Neuronal cell bodies are clustered in a series of interconnecting ganglia, which run along the entire length of the gastrointestinal tract. Closely associated with neurons, glial cells offer mechanical support and also have important roles in gut homeostasis and neurotransmission [2]. All neurons and glia of the ENS develop from neural crest-derived cells, most of which arise from the vagal region of the neural tube, migrating into the gut during weeks four to seven of human gestation [3,4]. Studies in animal models have shown that sacral-level neural crest and Schwann cell precursors also migrate into the gut and form the ENS [5,6], however, to what degree they contribute to the final ENS population in humans is yet to be determined. 

Hirschsprung disease (HSCR), also known as congenital aganglionosis, is characterised by the absence of the ENS in variable segments of the distal bowel. HSCR results from the failure of neural crest cells to appropriately migrate, proliferate or differentiate, leading to the development of an aganglionic zone. Clinically, the uncoordinated and tonic muscle contraction of the gut in HSCR results in functional obstruction of a variable length of colon, which may be fatal if left untreated [7]. Phenotypes of HSCR depend on the length of affected bowel. Short-segment disease, defined as aganglionosis to the junction of the descending colon and sigmoid colon, accounts for 60–85% of patients; long-segment disease (10% of patients) is characterised by aganglionosis proximal to the sigmoid colon with ganglion cells present in the colon; and total colonic aganglionosis is defined as aganglionosis of the entire colon extending into the terminal ileum (3–12%) [8,9,10,11]. Total intestinal aganglionosis is exceedingly rare, occurring in less than 1% of patients with HSCR, and is the most severe form of the disease [12]. 

Surgical resection of the aganglionic bowel and anastomosis of the proximal normoganglionated bowel to the anorectum is the mainstay management of HSCR to prevent morbidities and mortality associated with colonic obstruction. However, post-operative complications, including faecal incontinence (48%), constipation (30%), and enterocolitis still contribute to significant morbidity for the child [13,14,15,16]. Hirschsprung associated enterocolitis (HAEC) typically occurs in 25–35% of HSCR patients and manifests as fever, diarrhoea and abdominal distension [17]. Bowel dysmotility, immune dysfunction and the microbiome are thought to underline the disease process of enterocolitis [18,19,20], however, the exact mechanisms are unclear. Changes in ENS structure and innervation have also recently been implicated [21,22]. 

Diagnosis of HSCR occurs via rectal biopsies, which demonstrate the absence of enteric ganglia, and the presence of hypertrophic extrinsic nerves [23]. A transition zone typically occurs between the “normal” and aganglionic regions, with reduced density of enteric ganglia [24]. Prior to resection and subsequent anastomosis, intraoperative frozen sections are usually histologically evaluated by experienced pathologists to avoid “transition zone pull-through” (TZPT), where anastomosis occurs in the transition zone rather than the normoganglionated region. With an incidence of 14–18%, incomplete resection of the transition zone has been attributed to many post-operative complications, including obstruction, dysmotility and HAEC [14,25,26,27,28]. The transition zone is histologically characterised by partial aganglionosis of the bowel circumference, myenteric hypoganglionosis, and hypertrophy of submucosal and extrinsic nerves [29]. However, the specific criteria on how this is identified and interpreted by different hospital centres varies greatly. For example, the density of enteric ganglia can vary greatly, depending on how the tissue is processed and handled prior to fixation [30,31]. Many studies have also investigated the components of ganglia, reporting on differences between enteric neuron to glia ratios [31,32,33]. However, the total proximal segment of the resected bowel is not consistently evaluated, thereby affecting interpretation of outcomes [25]. This is in part due to the lack of formal protocols for surgical reporting and pathological evaluation, with recommendations from the American Pediatric Surgery Association (APSA) Hirschsprung Disease Interest Group only being published in 2019 [34]. Adding to this, there appears to be much variability in the ENS density and composition through the “normal” colon [30,32,33,35]. Despite strict intraoperative assessment of frozen sections and examination of full thickness proximal donut specimens prior to anastomosis, our institution continues to report post-operative complications, including enterocolitis. It is possible that TZPT is responsible for some of these outcomes, however, the underlying causes of many of these complications remain unknown. 

In this study, we aim to simplify the quantification of enteric ganglion density using the circular muscle unit as an unbiased measurement tool. By examining enteric ganglia relative to muscle units, variability from processing methods should have a limited impact on neuronal density calculation. To our knowledge, this study would be the first to investigate the changes in ganglion density in this manner. We hypothesise that the transition zone may be defined by reduced ganglion density within muscle units, thereby affecting bowel motility. 

## 2. Materials and Methods

### 2.1. Enrolment of Patients and Surgical Procedure

Ethics approval was obtained by The Royal Children’s Hospital Research Ethics Committee (HREC 38262). Between September 2019 and April 2021, 13 consecutive patients underwent the pull-through procedure for Hirschsprung disease at the Royal Children’s Hospital, Melbourne, Australia, and were included in this study. Twelve patients underwent Swenson transanal pull-through procedure and one patient with total colonic aganglionosis had a Duhamel pull-through (Table 1). 

For all patients, intraoperative frozen sections were used to identify the ganglionated bowel. During surgery, histological assessment of the bowel at our institute involves the use of both serial intestinal biopsies as well as a “doughnut” biopsy taken during bowel transection. For each patient, serial biopsies (a minimum of 3) were first taken to determine the location for anastomosis, one as the bowel’s calibre changes at the site of the megacolon, one 2 cm proximal to this, and one 2 cm distal to this site. If the most proximal biopsy was identified to be ganglionated and with an absence of hypertrophic nerve bundles, then no further biopsies would be taken. However, if the most proximal biopsy did not have any enteric ganglia, further biopsies proximal to this site would be taken. Once the ganglionated region of gut was established, a circumferential doughnut of tissue was taken 2 cm proximal to this site, and the bowel resected at this location. This proximal doughnut of tissue was sent for histological assessment using haematoxylin and eosin (H&E) and 1% Toluidine Blue (in 30% alcohol) staining. All the biopsies and the doughnut section from each patient were examined by an experienced pathologist during the surgery, and anastomosis surgery only took place if all tissue was determined to be in the normoganglionated gut and out of the transition zone [36]. Examination of the additional doughnut section is not routine in all hospitals, and at our institute, it is used to confirm the absence of partial ganglionosis and avoid transition zone pull-through. In total, patients experienced between one to three procedures: five patients only had a single pull-through surgery, while eight required a stoma prior to pull-through surgery. Complications and management strategies for these patients were found retrospectively using clinical notes, with the follow-up period ending in May 2021. Only complications identified following the completion of the final procedure were included. Two patients who still had an open stoma at the conclusion of the follow-up period were excluded from follow-up analysis. 

Data was also obtained from a single control patient, a three-year-old boy who suffered from neonatal necrotizing enterocolitis as a newborn. A stoma was made at that time and when the stoma was closed, we found a segmental stricture secondary to neonatal necrotizing enterocolitis located at sigmoid colon. The strictured colon was 10 cm in length and it was resected and processed in the same fashion as our HSCR patient tissue.

### 2.2. Tissue Processing and Immunohistochemistry

Following the operation, a full-thickness longitudinal strip (width = 5 mm, length = entire length of the resected tissue) was taken from the fresh specimen and was stored in cold PBS for a maximum of two hours, until further processing. The Swiss roll method was adapted for continuous examination [37]. The strip was divided into sections for ease of rolling, with the proximal end of the tissue rolled into the centre. On average, tissue from each patient was divided into 3.8 rolls (range from 1 to 6 rolls, depending on the resected bowel length). The specimen was fixed in 10% formalin overnight and then embedded into paraffin. Twenty slices were cut from each roll at 6 µm thickness along the longitudinal axis of the gut. Due to intraoperative histological evaluation for correct site of anastomosis, we were not able to include the most proximal 0.5–1 cm bowel segment in the current study. 

After deparaffinization, the specimens were immersed in 0.1 M trisodium citrate (pH 6.0) antigen retrieval solution at 60 °C overnight. The samples were then incubated with primary antibodies at 4 °C overnight and secondary antibodies at room temperature for 2 h. The primary antibodies used were mouse anti-HuC/D (Hu; Molecular Probes, Eugene Oregon, USA) and rabbit anti-S100B (1:800, DAKO, Agilent, Santa Clara, CA, USA). Secondary antibodies were donkey anti-mouse Alexa 594 (1:500, Molecular Probes), donkey anti-rabbit Alexa 488 (1:800, Molecular Probes), and DAPI. 

### 2.3. Image Analysis and Quantification 

For investigation of the muscle to ganglia ratio, the entire length of the resected tissue preparation from each patient was examined under a fluorescent widefield microscope (Zeiss, Oberkochen, Germany) and scored. The number of muscle units between two adjacent ganglia was counted, and the average muscle unit to ratio was calculated across three non-consecutive slides for each patient. A ganglion was defined as a cluster of at least two Hu+/DAPI+ neurons. For initial identification of muscle units and further investigation of individual ganglia, images were taken on an LSM880 confocal laser scanning microscope (Zeiss). For characterisation of individual circular muscle units, 10 randomly chosen images exhibiting a total of 23 muscle units from the proximal tissue segments were used. Images were taken with a 10× objective in order to capture the full radial thickness of the muscle (Figure 1). Measurements were conducted using Zen software (Zeiss). For glia–neuron ratio analysis, ganglia were imaged using a 40× objective. A minimum of three ganglia were captured from the most proximal tissue available (the “normal zone”), and three from the most distal ganglionated region (the “transition zone”). The numbers of DAPI+ nuclei, HuC/D+ neurons and S100B+ glia were manually counted using Image J (NIH). Areas stained without a clear DAPI+ nucleus were not included in cell counts. The average number of DAPI+ cells per ganglion, the proportion of Hu+/DAPI+ cells and the ratio of S100B+/Hu+ cells were then calculated in each region. 

### 2.4. Statistical Analysis 

Unless otherwise stated, all data are presented as mean ± SEM. All data were analysed using GraphPad Prism and were considered significant if *p* < 0.05.

## 3. Results

### 3.1. Patients Experienced Post-Operative Enterocolitis despite Intraoperative Histological Assessment of the Proximal “Doughnut” Biopsy

Tissue and clinical data were collected from 13 consecutive HSCR patients who underwent the pull-through procedure at Royal Children’s Hospital (Table 1), as well as a single control patient. Due to the loss of structural integrity of tissue following processing, Patient 10 was not included for analysis. Patient 13 was also excluded from analysis as he had total colonic aganglionosis, and hence, could not be compared to the other patients. Of the remaining 11 patients, eight were male (73%), and three patients were female (27%). All underwent Swenson pull-through (100%), with the average age at operation being 8.5 months (range 3–27 months). A range of Hirschsprung phenotypes were treated, with four patients (36.3%) suffering from short-segment disease and seven patients (63.6%) with long-segment disease. Two of these patients were syndromic as shown in Table 1. Seven patients also had either an ileostomy or colostomy prior to pull-through surgery (64%, Table 1). 

The follow-up period of this study ended in May 2021, with a mean follow-up time of 10 months (range 2–17 months) from the date of their final surgery (stoma closure if an ostomy was performed). At the conclusion of the follow-up period, one patient was unable to be evaluated as she still had an ileostomy (Patient 11). Half of the remaining 10 patients (50%) did not experience enterocolitis or constipation postoperatively. Whether or not these patients experienced faecal incontinence was difficult to assess as they had not yet been weaned from nappies. Five patients (50%) experienced at least one episode of enterocolitis. Of these patients, 80% had long-segment disease. Patient 8, who had short-segment disease and suffered from post-operative enterocolitis was subsequently found to have segmental aganglionosis (also known as “skip segments”). This was only identified after investigation of resected tissue using immunohistochemistry. Most patients (80%) with post-operative enterocolitis were managed with intermittent normal saline (0.9% sodium chloride) washes, with one patient having additional botulinum toxin injections. 

### 3.2. Identification and Definition of Individual Circular Muscle Units

To investigate the ratio of muscle units to enteric ganglia, we first defined and characterised circular muscle units in our resected patient bowel samples. A muscle unit was defined as a cluster of fibres that can be readily separated from adjacent clusters. Each circular muscle unit exhibited a depression at the border, referred to as a “junction” (Figure 1). As the outer border of muscle units was not consistently even, and some units appeared to have shallow depressions, we therefore set specific criteria, and the minimum depth of the junction valley was defined to be 30 µm. The width of each muscle unit was measured at 50 µm from the edge closest to the longitudinal muscle (Figure 1), and the mean was found to be 249 ± 62 µm. 

### 3.3. Muscle Unit to Ganglion Ratio Was Higher in the Transition Zone Compared to Proximal Colon

To examine the muscle unit to ganglion ratio, the entire length of the resected tissue was sectioned along the longitudinal axis and scored following immunohistochemistry. For the single control patient, the number of muscle units between individual ganglia was counted along the whole length of resected tissue, and the average ratio was calculated to be 1.03. 

For HSCR patients, we also examined the entire length of the resected tissue. Following immunohistochemistry, changes in the density of enteric ganglia along the longitudinal axis could be observed (Figure 2). A number of key criteria have previously been described in the transition zone of HSCR patients [29]. In our longitudinal colon sections, the presence of a myenteric hypertrophic nerve bundle (>40 µm) was used to identify the transition zone, as described by Subramanian et al. [38]. This criterion was employed as the authors used similar patient samples as our current study (HSCR patient tissue resections from primary surgery sectioned along the longitudinal axis). Hypertrophic nerve bundles could easily be identified using S100B staining (Figure 2B,B’); they were sometimes similar in appearance to enteric ganglia, however, lacked HuC/D+ neurons. To prevent sampling error, all data were examined across three non-consecutive sections. 

In the proximal tissue segment, oral to the first hypertrophic nerve bundle, the muscle unit to ganglia ratio was found to be 1.29 ± 0.07. Although we could not perform statistical analysis, these values were very similar to data from the normal bowel of the control patient. We, therefore, designated this proximal tissue as the normal zone. 

All tissue between the most proximal hypertrophic nerve bundle and the most distal ganglion was classified as the transition zone (Figure 2). The muscle unit to ganglion ratio in the transition zone was found to be 1.9 ± 0.2, which was significantly higher compared to the proximal, normal tissue segment (*n* = 11, Figure 2D). We further divided patients based upon the severity of aganglionosis (Table 2). For patients with long-segment HSCR (*n* = 7), we identified a significant difference between the normoganglionated colon and transition zone; however, the two regions were not statistically different for short-segment patients (*n* = 4). As only four patients with short-segment disease were analysed in this study (due to exclusion of Patient 10), further investigation is warranted. 

### 3.4. Ganglia Composition Is Unchanged in the Transition Zone

We examined the properties of individual ganglia in the proximal normal zone (NZ) and transition zone (TZ) in sections from HSCR patient tissue. Immunohistochemistry was performed against the pan-neuronal marker, HuC/D (Hu), and enteric glia were labelled with S100B (Figure 3). We examined the size and proportions of neurons and glia within each ganglion. There was no significant difference between the size of ganglia (number of DAPI+ cells/ganglion) in the transition zone (22.2 ± 4.2, *n* = 8) compared with the normal zone (23.6 ± 2.7, *n* = 8, *p* = 0.71, paired *t*-test), although these are likely to be an under representation of the true values as many ganglia were not completely captured by the field of view. The proportion of Hu+ neurons (out of DAPI+ cells in the ganglion) was not significantly different between the two regions (NZ: 29.8 ± 2.1%; TZ: 28.0 ± 4.0%; *p* = 0.64, paired *t*-test), and the ratio of glia to neurons was also not different between the regions (NZ: 2.8 ± 0.6; TZ: 3.2 ± 0.8; *p* = 0.56, paired *t*-test). There were also a number of nuclei present in the ganglia that were not immunoreactive for either Hu or S100B, and the identity of these cells remains unknown. Our data show that, although the ganglia are more sparsely located, the components of the ganglia are remarkably similar in the transition zone when compared with the normoganglionated region. 

### 3.5. Patients That Develop Enterocolitis Have Reduced Enteric Ganglion Size

Hirschsprung-associated enterocolitis is a common complication that develops in HSCR patients [17]. While changes in immune function and microbial dysbiosis are important, recently, changes in ENS structure and function have also been implicated in HAEC [21,22]. We, therefore, also investigated differences between the proximal normoganglionated tissue of patients that were shown to develop HAEC, with those that did not. The muscle unit to ganglia ratio was not significantly different between patients who experienced post-operative enterocolitis (1.31 ± 0.1, *n* = 5) and patients who had no reported post-operative complications (1.29 ± 0.1, *n* = 5; *p* = 0.90, Students *t*-test, Figure 4). In addition, we compared individual ganglia in HAEC patients with those of HSCR patients who had no complications. The size of ganglia was significantly smaller in HAEC patients when compared with non-HAEC patients (Figure 4), and again, this is likely to be an under-representation as many ganglia were not completely captured by the field of view. However, there were no significant differences between the proportion of neurons or the glia to neuron ratio (Figure 4). 

## 4. Discussion

The transition zone has been extensively examined via histological means to reduce rates of transitional pull-through resection. However, there appears to be much variability in tissue handling and measurement of various parameters. Previously, studies had sought to characterise the length of the transition zone, however, this had been limited to small numbers of patients with short-segment HSCR [29]. More recently, it was identified that the length of dysganglionosis is highly variable, ranging from 0.3 cm to 22.9 cm in patient tissue, and therefore, this study advocated for the analysis of individual patient samples using circumferential tissue sections, rather than relying on a specific length to identify the transition zone [25]. However, in addition, the distribution of ganglia in the transition zone is now understood to be uneven, resulting in a ‘leading edge’, which contributes to misidentification as the normoganglionated region and resulting TZPT [39]. Previous studies have shown that quantitative assessment, such as the measurement of submucosal nerve trunks or interganglion distance, has been proposed to prevent the morbidity associated with TZPT [38,40]. However, agreement on parameters has not been achieved and some measurement parameters are not realistic in the clinical setting of assessing intraoperative biopsies, being either too time-consuming or prone to variability. In our current study, we used the criterion of the presence of a myenteric hypertrophic nerve bundle with a diameter >40 µm to indicate the transition zone in our patient tissue. The diameter of approx. 40 µm was previously described as the upper limit of the thickness of a “normal” myenteric nerve bundle in longitudinal gut resections [38]. Although most studies have investigated submucosal nerve hypertrophy [23,29], and have previously stipulated the identification of two hypertrophic submucous nerves within a single ×400 magnification field of view, we used the more stringent criteria of a single hypertrophic nerve bundle identified in the myenteric plexus, which has also been previously described [38]. This ensured we did not underestimate the transition zone in our samples, however, comparisons of the equivalent regions to other studies could be more difficult. 

Our study is the first to use the muscle unit as a distinct, unbiased landmark for the quantification of enteric ganglia density in relation to their spatial distribution. We believe that this method reduces sampling bias, which is a significant challenge in current methods of intraoperative frozen section analyses [38]. Our results demonstrate the normal muscle unit to ganglia ratio to be approximately one. In the transition zone of long-segment HSCR patients, this ratio was significantly increased, indicating a reduction in the ganglia density. Interestingly, while increased muscle unit to ganglion ratio was observed for some short-segment HSCR patients, we did not find a statistically significant difference to the normoganglionated region. This could be due to clinicopathological differences between the two phenotypes. Variation in genetic profiles have been noted, with mutations in *SEMA3C*, important for axonal guidance and neural crest cell migration, being more common in long-segment compared to short-segment disease [9,41,42,43]. In contrast to short-segment disease, submucosal nerve hypertrophy may be limited or absent in rectal biopsies of long-segment HSCR, contributing to longer time to diagnosis of 11–14 days compared to 2–3 days for short-segment disease [8,9,23]. Furthermore, Solari et al. have demonstrated a lack of interstitial cells of Cajal and markedly reduced NADPH-positive nerve trunks in smooth muscle layers of total colonic aganglionosis patients. This may contribute to the poorer outcomes associated with long-segment disease and its association with HAEC [10,15,44]. 

Our novel approach simplifies quantitative measures and reduces the variability resultant from different tissue processing techniques. However, our study is currently limited by sample size with only one control specimen, and further investigation is required before this can be integrated into intraoperative analysis for transition zone identification. Thus far, we have assessed the muscle unit to ganglion ratio for the circular muscle, as the tissue was sectioned along the longitudinal axis of the gut in order to capture data from the entire length of the resected bowel segment. Further investigation examining the longitudinal muscle using transverse sections will help build strict parameters for adopting the muscle unit as a tool for quantifying ganglia density. In the current project, analysis of the muscle unit to ganglion ratio was performed following fixation and immunohistochemistry to provide further characterisation of the patient tissue. A faster method of identifying ganglia and muscle units is needed before this standard can be adopted for routine analysis during pull-through surgery. H&E and toluidine blue staining on donut sections of resected gut or biopsies taken to identify the presence of ganglia, can be used. Current protocols at our institution involve intraoperative examination of a circumferential donut biopsy of tissue from the proximal-most resected gut using H&E staining, which is given to an experienced pathologist to examine whether this tissue is part of the transition zone, using histological features as previously described, such as the presence of the hypertrophic nerve bundle [29]. Identification of muscle units can be integrated into this process or used as an adjunct procedure to ensure correct identification of the normoganglionated colon. 

Interestingly, further analysis of individual ganglia in our study revealed that while the density of ganglia in the transition zone was sparser, the composition of ganglia was remarkably similar to that of ganglia in the proximal, normal region. Previous studies have reported that the ratio of enteric glia to neurons is disrupted in the transition zone [31,32,33]. However, there is also variability of ganglia in the healthy, normal colon. In the myenteric plexus, this has been quoted across a range of 3.7 to 9 and can differ between different segments of the colon [31,32,33]. More recently, Graham et al. visualised normal colon in three-dimensions and found that the ratio of glia to neurons within myenteric ganglia was lower in the left colon (2.5) when compared with the right colon (4.2) [35]. It is possible that changes in glia to neuron density through different regions of the control colon may account for the variability that we observed and the lack of a significant difference. 

Our patient outcomes demonstrated that post-operative enterocolitis persists despite routine intraoperative analysis of frozen sections and full thickness proximal donut specimens. Limited by a small sample size and having a mean follow-up period of 10 months, our study recorded a 50% rate of post-operative enterocolitis. This is greater than findings of Thomson et al. in their systematic review, which revealed an incidence of 10% to 45% [45,46]. It is recognised that patients with long-segment disease have poorer outcomes compared to patients with short-segment phenotypes, experiencing higher rates of incontinence and enterocolitis [10,15]. This may account for our poorer outcomes given a high proportion of long-segment disease in our patient cohort (80%). Although exact mechanisms remain unclear, emerging concepts have implicated impaired gut barrier and immune function in the pathogenesis of enterocolitis. Whilst initial studies have linked HAEC with infection of *Clostridium difficile* and rotavirus, more recent studies have suggested an altered microbial community may result in impaired gut homeostasis and barrier function [18,19]. Chronic dysmotility and stasis of faecal matter feeds into dysbiosis and aberrant inflammatory processes and, hence, highlight the importance of identification of the transition zone to prevent TZPT and subsequent enterocolitis [26]. 

Our study shows that comparison of ganglia density using the muscle unit to ganglion ratios in the proximal regions of resected gut of patients experiencing post-operative enterocolitis to those who did not reveal any significant difference between patient cohorts. Coupled with our institute’s current protocols for identifying the transition zone, this suggests that TZPT may not be the cause of enterocolitis in our patients. Interestingly, our study identified that HAEC patients had smaller ganglion size in their normoganglionated region compared to control patients. These smaller ganglia appear to be different to “myenteric hypoganglionosis”, where ganglia consisting of single cells or doublets have been identified in the transition zone of HSCR patients [29]. Instead, our data show that there appears to be fewer large ganglia in HAEC patients. Within the ganglia, there were no significant differences between the proportions of neurons or glia; however, the decreased ganglia size would indicate that there is an overall decrease in the density of enteric neurons and glial cells. Overall, this suggests that more subtle changes in the ENS may underly the development of HAEC. Reductions in enteric cholinergic innervation of the gut mucosa and serotonin production have been found in HAEC patients [21,22,47]. It is now recognised that the ENS communicates with the gut immune system and has an important role in immune activation and suppression [48,49,50]. There are many pathways involved in this communication [49,51], and also reciprocal interaction where the immune system influences ENS function [52]. Recent data shows that enteric glia also play important roles in this exchange [53,54]. Whether loss of other subtypes of neurons or specific enteric glial cells contribute to HAEC remains to be investigated further. 

## 5. Conclusions

Correctly and quickly identifying the transition zone during HSCR pull-through surgery is important for reducing post-operative complications. Our study shows that the correlation between the ratio of muscle units to enteric ganglia may be developed as a quick and reliable method for identifying normal versus hypoganglionosis in the transition zone. Further investigation to examine longitudinal muscle units, and comparison of smaller tissue samples, for example doughnut samples or biopsies, will help develop this method for intraoperative use. In addition, we found that subtle changes in individual ganglia may contribute to the development of post-operative complications, including enterocolitis. This most likely reflects subtle defects in ENS development, and thus impacts on gut function. Further investigation of the specific populations of cells and signalling mechanisms involved will help understand how these subtle changes in ENS may lead to dramatic imbalances in gut function. 

## Figures and Tables

**Figure 1 biomolecules-12-01101-f001:**
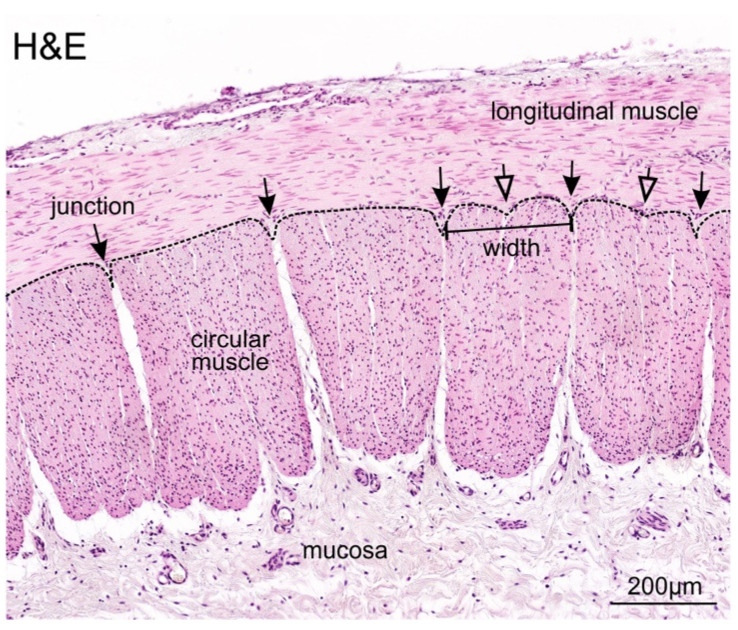
Representative H&E image of the tissue architecture in the transition zone of a long-segment HSCR patient. Circular muscle units are outlined (dotted line), separated at junctions (arrows) where there is a clear depression. Shallow depressions (open arrows) were also present, but were <30 µm in depth, and thus, were identified as being located within muscle units.

**Figure 2 biomolecules-12-01101-f002:**
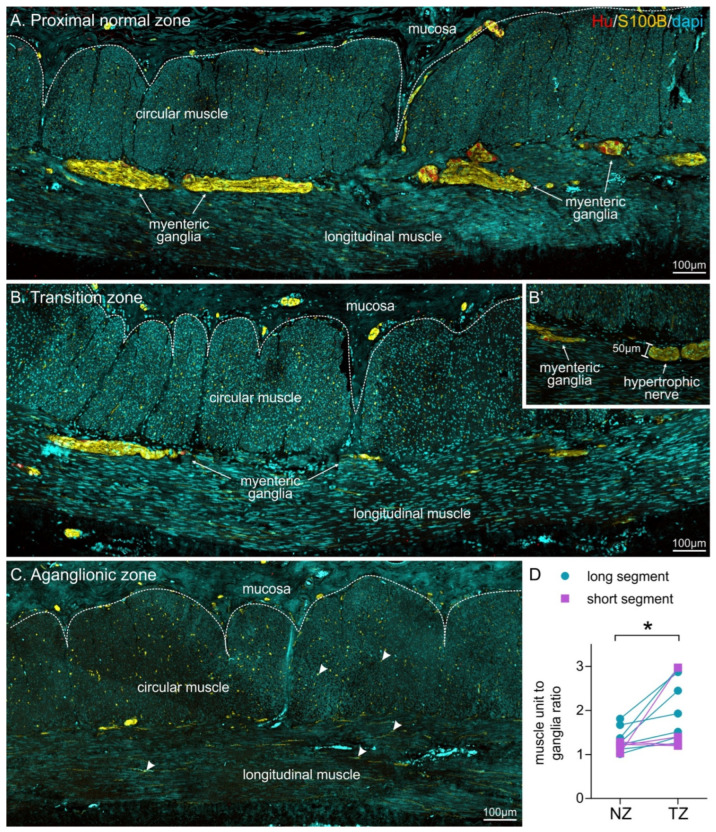
Overview of the distribution of myenteric ganglia in relation to circular muscle units in the proximal normal zone (**A**, NZ), the transition zone (**B**, TZ), and the aganglionic zone (**C**) in longitudinal sections of HSCR patient tissue following immunohistochemistry against HuC/D (Hu), S100B and DAPI. Ganglia are more sparsely located in the transition zone, and no ganglia are visible in the aganglionic zone, although individual glial cells are present (some highlighted by arrowheads). (**B’**) shows a hypertrophic nerve bundle, which could be identified using S100B immunohistochemistry, but lacked neuronal cell bodies. Hypertrophic nerve bundles had a minimum width of 40 µm. (**D**) quantification of muscle unit to ganglion ratio in both long- and short-segment patients (* *p* = 0.012, *n* = 11, paired *t*-test).

**Figure 3 biomolecules-12-01101-f003:**
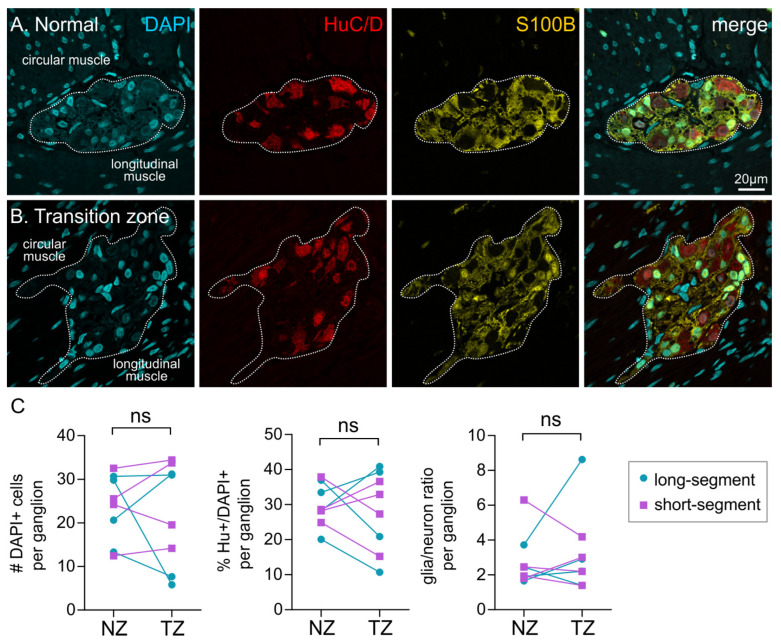
Neuron–glia composition of enteric ganglia in the normal vs transition zones. (**A**,**B**) Representative images of ganglia from the normal (**A**) and transition zone (**B**), with immunohistochemistry performed against DAPI, HuC/D (Hu) and S100B. (**C**) Quantification of the size of ganglia (#DAPI+ cells/ganglion, *left*), the proportion of neurons per ganglion (%Hu+/DAPI+, *middle*), and the ratio of glia/neurons per ganglion (*right*) in four long-segment and four short-segment HSCR patients. ns: *p* > 0.05, paired *t*-test for all.

**Figure 4 biomolecules-12-01101-f004:**
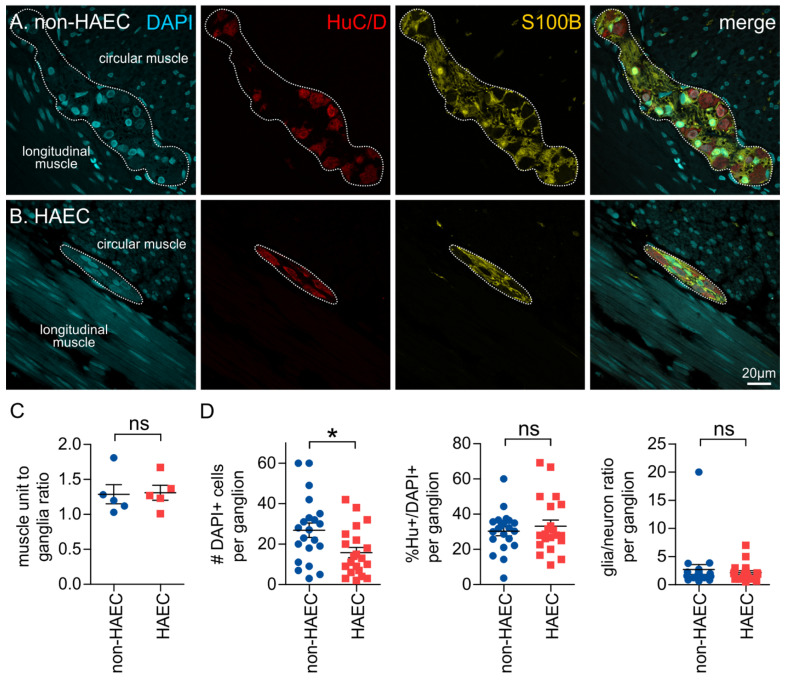
Comparison of ganglia from the proximal normoganglionic tissue in non-HAEC and HAEC patients. (**A**,**B**) Representative images of ganglia from the patient groups, with immunohistochemistry performed against DAPI, HuC/D and S100B. Note: (**C**) Quantification of the muscle unit to ganglia ratio (ns: *p* > 0.05, Students’ *t*-test). (**D**) Quantification of the size of ganglia (#DAPI+ cells/ganglion, *left*), the proportion of neurons per ganglion (%Hu + /DAPI+, *middle*), and the ratio of glia/neurons per ganglion (*right*). non-HAEC: *n* = 21 ganglia, *N* = 4 patients; HAEC: *n* = 21 ganglia, *N* = 3 patients. * *p* = 0.0158, ns: *p* > 0.05, Students’ *t*-test.

**Table 1 biomolecules-12-01101-t001:** Patient demographics and severity of disease.

Patient	Sex	Age at Operation (months)	Phenotype	Procedure(s)	Follow-Up Duration (Months from Date of Last Procedure)	Complication
**HSCR**						
Patient 1	Female	6	Long—proximal transverse colon	3: Ileostomy + Swenson + stoma closure	17	none
Patient 2 *	Male	6	Long—proximal descending colon	3: Ileostomy + Swenson + stoma closure	11	none
Patient 3	Male	7	Long—distal transverse colon	3: Ileostomy + Swenson + stoma closure	11	Three episodes of enterocolitis
Patient 4	Male	4	Short	3: Ileostomy + Swenson + stoma closure	12	none
Patient 5	Male	5	Long—distal descending colon	1: Swenson (no stoma)	15	Three episodes of enterocolitis
Patient 6	Male	3	Short	1: Swenson (no stoma)	10	none
Patient 7	Male	21	Short	3: Ileostomy + Swenson + stoma closure	2	none
Patient 8	Male	4	Short—rectal sigmoid (skip)	2: Colostomy + Swenson	7	Several episodes of enterocolitis
Patient 9	Male	6	Long	1: Swenson (no stoma)	6	Three episodes of enterocolitis
*Patient 10*	*Male*	*3*	*Short*	*1: Swenson (no stoma)*	*6*	*Constipation*
Patient 11	Female	27	Long—transverse colon	2: Colostomy + Swenson (ileostomy formed following pull-through)	-	Stoma not yet closed
Patient 12	Female	5	Long—descending colon	1: Swenson (no stoma)	13	One episode of enterocolitis
*Patient 13 #*	*Male*	*27*	*Total colonic aganglionosis*	*2: Ileostomy + Duhamel*	*-*	*Stoma not yet closed*
**Control**						
Patient 1	Male	36	Neonatal necrotizing enterocolitis	2: Ileostomy + stoma closure with resection of strictured bowel	7	N/A (no follow-up)

* Patient with Waardenburg type IV. # Patient with Down syndrome. Patients 10 and 13 were excluded from tissue analysis.

**Table 2 biomolecules-12-01101-t002:** Muscle unit to ganglion ratios.

HSCR Type	Proximal/Normal	Transition Zone	Statistical Analysis
All patients (pooled)	1.29 ± 0.07	1.93 ± 0.22	* *p* = 0.012; *n* = 11
Long-segment	1.35 ± 0.1	2.06 ± 0.3	* *p* = 0.017; *n* = 7
Short-segment	1.19 ± 0.05	1.70 ± 0.43	ns: *p* = 0.36; *n* = 4

* denotes significant *p*-values.

## Data Availability

All data from this study can be made available upon request. Please contact the corresponding authors.

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
