# Peer review of "A Novel Method for Identifying the Transition Zone in Long-Segment Hirschsprung Disease: Investigating the Muscle Unit to Ganglion Ratio"

_biomolecules, 2022, doi:10.3390/biom12081101_

Round 1

Author Response

REVIEWER 1:

The study by Yang et al is a clearly written, easily understood quantitative analysis of enteric neural tissue and musculature in the proximal ganglionic segments of bowel resected from patients with Hirschsprung disease (HD). The figures nicely complement the text. The hypothesis and results have potential implications for how patients with HD might be managed clinically, although additional research would be required before any changes to the standard of care can be recommended. The following concerns should be considered:

RESPONSE: We thank the reviewer for their careful review of our work. Please find the amended manuscript attached below. 

1. Introduction, lines 90-95: Implicit in this text (“... additional post-operative criteria are required ...”) is an assumption that transition zone pullthrough (TZPT) is the reason for post-operative enterocolitis (HAEC) in their patient population. This assumption permeates much of the literature in this field and the authors need to be careful to acknowledge that insufficient well- controlled data exist to conclude that TZPT, regardless of what features are used to define transition zone, increases the risk for post-operative HAEC.

RESPONSE: We thank the reviewer for this excellent point. We have modified the last 2 paragraphs of the Introduction to acknowledge that many factors could contribute to HAEC, not just TZPT.

2. Methods:

2A. It appears as though most or all of the patients in this study had two-stage procedures (ostomy followed later by pullthrough) and that some or all may have had a third procedure to take down the ostomy and re-anastomose more proximal bowel. Some this this detail should be added to the Methods section and or Table 1. It is important to include in Table 1 and all the data analysis how long patients were followed after bowel continuity was completely established, as TZPT should not have any impact until a patient’s ostomy is taken down. Either only episodes of HAEC that occurred after continuity was fully established should be counted in the data presented in Figure 4 and/or the data should be parsed to clearly show what percentage, if any, of the post- op HAEC occurred after ostomy takedown.

RESPONSE: Investigation of HAEC and post-operative complications were identified following the final anastomosis surgery and anything that was identified in the interim time between surgeries were not included. We apologise that this was not more clear, and have included this information in the Methods (section 2.1), as well as the follow-up time for each patient in Table 1. Out of our final 11 patients analysed, 7 had an ostomy. Patients had a range between a single procedure (Swenson pull-through only) to 3 procedures (ostomy + pull-through + stoma closure). This information has now been added to Table 1, we apologise for not including it earlier.

2B. The text suggests that resection was performed, at least initially, 2 cm proximal to a leveling biopsy that contained ganglion cells and that doughnut sections were used intraoperatively to exclude TZPT at the proximal margin. This is less than the 5 cm advocated in some papers (cited in the manuscript) for short-segment HD. How often, if ever, was TZPT encountered in the doughnut sections and additional bowel resected? If additional bowel was resected, was it included in the analysis? If not, it seems unfair to attempt to correlate post-operative HAEC with findings at the proximal end of a resection specimen, which does not represent the most proximal ganglionic bowel resected from the patient.

RESPONSE: For all surgeries, biopsies were first taken and examined to identify the normoganglionated gut. The doughnut section was then taken and examined after investigation of the biopsies. The doughnut section and gut resection was performed 2cm proximal to the oralmost levelling biopsy. We apologise for missing this information in our original manuscript. This has now been added to the Methods (section 2.1). This means that for the vast majority of patients, the distance between the visual gut diameter change at the onset of the megacolon and the resection location would be at least 4-5cm. Please note, this is the distance gauged during surgery.

Only if the doughnut section was confirmed to be normoganglionated would the anastomosis surgery take place. For all 13 patients in this study, the first doughnut section taken was confirmed to be normoganglionated, so no further sections were required. For the vast majority of cases in our centre, only a single doughnut section has been necessary, as it confirms information from the levelling biopsies. We apologise this information was not clearly stated and have expanded this information in the Methods (section 2.1). As a rough estimate, the doughnut section has been found to be in the transition zone for 1 patient in every 3-4 years.

2C. Table 1 should include the length of the ganglionic proximal segment in each resection specimen. Data analysis should include whether quantitative parameters correlated with this length, not just length of the aganglionic segment.

RESPONSE: Apologies but this information was not documented for each patient.

2D. Only one control specimen was included in this study. Although the authors acknowledge this limitation in the Discussion, this limitation deserves more emphasis. Given variability in quantitative data, which exist in the literature (cited in manuscript), it is unclear that data from one control is very meaningful. The authors might bolster the value of this control by assessing variability in the quantitative data obtained from counts in different parts of the control specimen, instead of providing just one value for muscle unit/ganglia of 1.03. What range of values might be obtained if the analysis was repeated in different areas? Did location (upstream or downstream) from the stricture make any difference?

RESPONSE: We agree with the reviewer that there are severe limitations with a single control patient. To minimise variation within samples, analysis of the muscle unit to ganglion ratio was performed for the entire length of the gut that was resected. This was performed for the control sample as well as all HSCR patient resections. We apologise that this method was not included in the manuscript, this has now been added to the Results (beginning of section 3.3). Therefore, the ratio of 1.03 represents the data gathered from the entire sample.

2E. It appears that no effort was made to correlate the findings from the longitudinal strip analysis with the frozen section or permanent section proximal margin histology, despite the fact that proximal margin doughnuts were obtained from every patient. From a practical standpoint it is important to know whether the abnormalities detected morphometrically by immunohistochemistry in some, but not all, of the patients were (or could have been) detected at the proximal margins of the original sections. The authors suggest this might be the focus of a future study, but should seriously consider adding this histological assessment to the current study. This could be accomplished simply by review of the existing H&E- and/or toluidine blue-stained sections by an experience pathologist.

RESPONSE: Unfortunately we are currently unable to correlate our findings on the muscle unit to ganglion ratio to the proximal doughnut sections as they were sectioned in different directions. Proximal doughnuts were examined as transverse sections while the resected tissue was sectioned along the longitudinal axis. Therefore, the information from the resected tissue correlates to circular muscle units, while the proximal doughnuts would only yield information about longitudinal muscle units. Further, for all patients, the proximal doughnut collected during resection was analysed and considered to be normoganglionated according to analysis by an experienced pathologist. Although we agree with the reviewer that it would be good to confirm our findings in these proximal doughnut sections, we no longer have access to this tissue.

2F. Methods, line 163-4: Were ganglia imaged with a 40x objection (400x magnification) or really at 40x magnification as stated?

RESPONSE: Apologies, they were imaged with a 40x objective. This has now been corrected in the Methods (page 5).

3. Results

3A. Patient 8 should be excluded from HAEC analysis since he had a skip area. One might argue that he should be excluded completely since the transition zone of a patient with a skip area may be different from classic HD.

RESPONSE: Patient 8 was very interesting as the “skip segment” were only identified after investigation of the resected tissue using immunohistochemistry. The patient was diagnosed with short-segment HSCR based on investigation of the biopsies and doughnut section. The skip segment was only 1cm in length, hence, we do not believe this has serious consequences on the tissue analysis. As HSCR patients with skip segments are very rare, it has been difficult to build a thorough understanding of their tissue morphology and clinical pathology. Hence, we do not agree that Patient 8 should be excluded from our analysis.

3B. Results, section 3.3., line 221-2: proximal bowel is defined as “oral to the first hypertrophic nerve”. First, how was a hypertrophic nerve defined? Second, the distance between large myenteric nerves in a single longitudinal section, even in the aganglionic segment, can be quite long and variable (see Figure 2C in the manuscript). Is it fair to use this landmark as an index for “normalization”?

RESPONSE: Hypertrophic nerves were identified using S100B staining. They had a minimum width of 40um. Apologies for not including this information, this has now been added to Results section 3.3. All tissue proximal to the most oral identification of the hypertrophic nerve bundle was considered to be “normal” tissue. This is based on previous publications where the presence of a hypertrophic nerve bundle was used as a landmark to identify the transition zone. Apart from identification of where the transition zone starts, the hypertrophic nerve bundles were not used for any further characterisations.

3C. The patient with total colonic aganglionosis and a TZ in the small intestine should probably be excluded given that no control data is available for small intestine. If patient’s 8 and 13 are excluded, is the muscle unit/ganglia ratio still significantly higher for TZ vs proximal?

RESPONSE: We agree with the reviewer that patients 13 should be excluded from the analysis. Table 1, Table 2, Figure 2 and Figure 3 have been updated with these results. There have been no further changes to our analysis and conclusions.

3D. Figure 2 needs scale bars.

RESPONSE: Apologies, we have now added scale bars to all panels.

3E. Figure 3 and related text. Many intraganglionic DAPI-positive nuclei are present which neither colocalize with HuC/D nor S100. What are these cells? Could they be S100- negative glial cells?

RESPONSE: This is an excellent question, we’re not sure about the identity of these cells. It is possible that they are S100B-negative glial cells, however, we did not have any success with Sox10 or GFAP immunohistochemistry, despite trying many different antigen retrieval methods. We have added a sentence about these cells in the Results (section 3.3).

3F. Figure 4B: The images resemble what has been termed “myenteric hypoganglionosis,” one of three primary H&E-features of TZ (along with partial circumferential aganglionosis and submucosal nerve hypertrophy) (see ref 29). This feature can and should be recognizable to an experience pathologist. It would be important to know if this was present in the corresponding doughnut sections from this or other similar patients (see concern 2a above) and to include discussion of the overlap with existing criteria for TZ in the Discussion.

RESPONSE: This is an interesting point. A number of ganglia from all HSCR patients were small in size, consisting of only single or doublets of dapi+ nuclei. However, these were not obviously more prominent in patients that went on to develop HAEC. The difference in ganglia size between patients appear to be due to a smaller number of large ganglia in HAEC samples. We have altered the graphs in figure 4D with individual data points for each ganglion to help illustrate this. We have also added a few sentences to the Discussion about this (page 13).

4. Discussion

4A. Discussion, line 299: The statement that the transition zone “had previously been documented to be no greater than 5 cm” is an oversimplification of what was reported. The study referenced concerned only short-segment HD. Long-segment HD is generally known to have longer TZ, as mentioned in Discussion sections of some of the cited publications by Kapur and colleagues.

RESPONSE: We apologise for this error, this section of the Discussion has been amended.

4B. Discussion, line 313-5: The idea that a single ganglion controls a single muscle unit because in an individual longitudinal section the average muscle unit to ganglia ratio was close to 1 ignores the fact that these sections only sample a tiny fraction of the circumference and that existing physiological data suggest that neurons (e.g. inhibitory neurons) project relatively long distances up- or downstream to muscle cells to regulate contractility.

RESPONSE: We apologise for this simplified statement. We had thought to introduce the idea that on average, it appeared that a ratio of a single ENS ganglion to a muscle unit was needed to have sufficient innervation. However, we agree with the reviewer that this is an oversimplification, as it doesn’t take into account the long projections of enteric neurons. We have removed references to this from the text.   

4C. Discussion, line 325: Replace “neural hyperplasia” with “submucosal nerve hypertrophy”.

RESPONSE: This has been amended.

Reviewer 2 Report

This is a very interesting and clinically-relevant paper that identifies a novel method for identifying the transition zone in Hirschsprung disease (HSCR). Although surgery for HSCR can be life-saving, a resection margin that involves transcition zone – the region of bowel between normal gut proximally and aganglionic distal intestine – is a cause of unsatisfactory postoperative outcomes. Unfortunately, current methods for identifying the transition zone are inadequate. Using a relatively small cohort of patients (n=12), the authors identify the ratio of muscle units to ganglia as a novel metric that may identify transition zone. They report that the ratio is close to 1 in normal intestine, and is reduced in the transition zone. Measuring this variable may be a valuable new approach for determining whether the resection margin is adequate. Overall, this is a novel and interesting study. There are, however, a few points that should be addressed.

In the introduction, line 62, the authors list some common long-term complications of HSCR, in which they include short bowel syndrome. This is incorrect. Short bowel syndrome results from resection of large amounts of small intestine, with resulting loss of nutritional autonomy. This is not a complication of surgery for HSCR in the overwhelming majority of cases.

The title and abstract imply that their metric is useful for identifying transition zone in general, but their data (Fig 2) shows that a significant difference is only seen in long-segment disease. The lack of difference in short-segment HSCR in this study may be due to lack of power (only 4 short-segment patients were studied). Nevertheless, the claim that the muscle unit to ganglia ratio can identify transition zone should be less broad and limited to long-segment disease. The title, abstract, and text should be updated to make clear this limitation.

While not strictly necessary, I would encourage the authors to validate their method via H&E staining on frozen sections of fresh tissue. This approach would be necessary for clinical translation, since surgeons require rapid information during operations to decide whether to extend the resection.

Author Response

REVIEWER 2:

This is a very interesting and clinically-relevant paper that identifies a novel method for identifying the transition zone in Hirschsprung disease (HSCR). Although surgery for HSCR can be life-saving, a resection margin that involves transcition zone – the region of bowel between normal gut proximally and aganglionic distal intestine – is a cause of unsatisfactory postoperative outcomes. Unfortunately, current methods for identifying the transition zone are inadequate. Using a relatively small cohort of patients (n=12), the authors identify the ratio of muscle units to ganglia as a novel metric that may identify transition zone. They report that the ratio is close to 1 in normal intestine, and is reduced in the transition zone. Measuring this variable may be a valuable new approach for determining whether the resection margin is adequate. Overall, this is a novel and interesting study. There are, however, a few points that should be addressed.

RESPONSE: We thank the reviewer for their careful review. Please find attached below the revised manuscript. 

In the introduction, line 62, the authors list some common long-term complications of HSCR, in which they include short bowel syndrome. This is incorrect. Short bowel syndrome results from resection of large amounts of small intestine, with resulting loss of nutritional autonomy. This is not a complication of surgery for HSCR in the overwhelming majority of cases.

RESPONSE: Apologies for this oversight, we have removed short bowel syndrome from this statement.

The title and abstract imply that their metric is useful for identifying transition zone in general, but their data (Fig 2) shows that a significant difference is only seen in long-segment disease. The lack of difference in short-segment HSCR in this study may be due to lack of power (only 4 short-segment patients were studied). Nevertheless, the claim that the muscle unit to ganglia ratio can identify transition zone should be less broad and limited to long-segment disease. The title, abstract, and text should be updated to make clear this limitation.

RESPONSE: Thanks for this very important point, we have made the changes to the title, abstract and text. We have modified the Discussion (2nd paragraph) to clarify this difference.

While not strictly necessary, I would encourage the authors to validate their method via H&E staining on frozen sections of fresh tissue. This approach would be necessary for clinical translation, since surgeons require rapid information during operations to decide whether to extend the resection.

RESPONSE: We thank the reviewer for this suggestion and we acknowledge that this would indeed be a highly valuable addition to the study. However, this would mean re-analysing all of our samples to identify the ganglia to muscle unit ratio after H&E staining and we would be unable to complete this in a timely manner. The aim of this study was to examine whether there were any underlying differences between the muscle unit to ganglion ratio in the transition zone of HSCR patients, which we believe we have been able to answer. We agree with the reviewer that the next step would be to examine the factors that would need to be in place for clinical translation of this finding. However, as such, it is beyond the scope of the current project.

Round 2

Reviewer 1 Report

The revised manuscript is much improved and addresses all but one of my major concerns with the original manuscript.  The concern that remains regards identification of the transition zone, which is described as follows:

"To identify the transition zone, the presence of a hypertrophic nerve bundle was used, as described previously [29]. Hypertrophic nerve bundles could easily be identified using S100B staining, with a minimum width of 40um (Figure 2B, B’)."

This description and the reference figure are flawed.  As indicated in reference 29, but misrepresented in the manuscript, the "40 micrometer" cut-off for nerve hypertrophy (a) regards submucosal, not myenteric, nerves, (b) requires 2 nerves >40 micrometers in diameter in one 400x field, and (c) stems from the original paper of Coe et al (Pediatr Dev Pathol 2012;15:30-38).  As I mentioned in my original review, it is unusual to encountered submucosal nerve hypertrophy in long-segment HD (TZ proximal to the splenic flexure), as opposed to short-segment HD, presumably because submucosal nerve hypertrophy gradually lessens with distance from the rectum.

The authors appears to have used their own novel criterion to identify the transition zone based on myenteric nerve caliber of >40 microns.  This fact should be made VERY clear and the limitations associated with use of this criterion should be discussed.  It must also be acknowledged that defining the TZ based on assessment of nerve hypertrophy in a single longitudinal section, is likely very inaccurate since nerves are not distributed uniformly around the circumference of the bowel.

Author Response

RESPONSE: We sincerely apologise, we had made an error with our citation, the 40um cut-off for nerve hypertrophy was identified by Subramanian et al., 2017 (reference #38 in manuscript). This study examined longitudinal sections of HSCR patient resected tissue from their primary surgery, and also compared control patient samples processed using the same methods. Their study found that the upper limit for the width of a “normal” myenteric nerve was 36um (mean = 27.8um), and therefore, nerves that were larger than this size were considered as hypertrophic. We realise that the majority of studies investigating hyertrophic nerve bundles have investigated submucous nerves, and hence, it could be difficult to draw comparisons between our study and others. However, we elected to go with the more conservative estimate for a single hyertrophic nerve in a field of view at 400x magnification, which was also indicated in Subramanian et al.’s study. This was conducted to ensure we did not underestimate the transition zone in our samples. To avoid data sampling errors, our analysis was conducted across 3 non-consecutive sections. We have added this information to our Results (page 7) and Discussion (page 12).

Round 3

Reviewer 1 Report

Thank you for addressing my concerns.  This is a well written and illustrated study.